# Endoscopic Ultrasound-Guided Through-the-Needle Biopsy: A Narrative Review of the Technique and Its Emerging Role in Pancreatic Cyst Diagnosis

**DOI:** 10.3390/diagnostics14151587

**Published:** 2024-07-23

**Authors:** Filipe Vilas-Boas, Tiago Ribeiro, Guilherme Macedo, Jahnvi Dhar, Jayanta Samanta, Sokol Sina, Erminia Manfrin, Antonio Facciorusso, Maria Cristina Conti Bellocchi, Nicolò De Pretis, Luca Frulloni, Stefano Francesco Crinò

**Affiliations:** 1Gastroenterology Department, Centro Hospitalar e Universitário de São João, Faculdade de Medicina da Universidade do Porto, 4200-349 Porto, Portugal; filipe.vboas.silva@gmail.com (F.V.-B.); tiagofcribeiro@outlook.com (T.R.); guilhermemacedo59@gmail.com (G.M.); 2Department of Gastroenterology, Post Graduate Institute of Medical Education and Research, Chandigarh 160012, India; jahnvi3012@gmail.com (J.D.); dj_samanta@yahoo.co.in (J.S.); 3Department of Diagnostics and Public Health, University of Verona, 37129 Verona, Italy; sokol.sina@aovr.veneto.it (S.S.); erminia.manfrin@univr.it (E.M.); 4Gastroenterology Unit, Department of Medical and Surgical Sciences, University of Foggia, 71122 Foggia, Italy; antonio.facciorusso@virgilio.it; 5Gastroenterology and Digestive Endoscopy Unit, University Hospital of Verona, 37134 Verona, Italy; mariacristina.contibellocchi@aovr.veneto.it (M.C.C.B.); nicolo.depretis@univr.it (N.D.P.); luca.frulloni@univr.it (L.F.)

**Keywords:** mucinous cystic neoplasm, EUS-FNA, pancreatic cancer, cystic fluid analysis, intraductal papillary mucinous neoplasm (IPMN), neuroendocrine tumor

## Abstract

Pancreatic cystic lesions (PCLs) pose a diagnostic challenge due to their increasing incidence and the limitations of cross-sectional imaging and endoscopic-ultrasound-guided fine-needle aspiration (EUS-FNA). EUS-guided through the needle biopsy (EUS-TTNB) has emerged as a promising tool for improving the accuracy of cyst type determination and neoplastic risk stratification. EUS-TTNB demonstrates superior diagnostic performance over EUS-FNA, providing critical preoperative information that can significantly influence patient management and reduce unnecessary surgeries. However, the procedure has risks, with an overall adverse event rate of approximately 9%. Preventive measures and further prospective studies are essential to optimize its safety and efficacy. This review highlights the potential of EUS-TTNB to enhance the diagnostic and management approaches for patients with PCLs. It examines the current state of EUS-TTNB, including available devices, indications, procedural techniques, specimen handling, diagnostic yield, clinical impact, and associated adverse events.

## 1. Introduction

Pancreatic cystic lesions (PCLs) represent a significant burden for health care systems. Their diagnosis is increasing, with a pooled prevalence of 8% in a recent meta-analysis [1] and an incidence of 12.9% in a population-based study over a period of 5-year follow-up [2], explained by incidental detection in cross-sectional imaging [3,4]. Other drivers may also have a role, as the growth in new diagnoses far outpaces the growth of imaging use [5].

The spectrum of PCLs comprises lesions with no malignant potential, like inflammatory pseudocysts and serous cystadenoma (SCA) that need no resection or surveillance (unless symptomatic) to malignant lesions, like solid pseudopapillary neoplasm and cystic neuroendocrine tumors (cNETs), that benefit from surgery. Between these categories lies the group of mucin-producing lesions like intraductal papillary mucinous neoplasm (IPMNs) and mucinous cystic neoplasm (MCN) with different grades of dysplasia that require surveillance or resection. The majority of incidental PCLs are neoplastic, mostly IPMNs that have risk of malignant transformation [2].

International guidelines [6,7,8] are based on knowledge of specific cyst types, but cross-sectional imaging alone has limited accuracy for PCLs subtyping and risk stratification [9,10]. Moreover, despite the use of endoscopic ultrasound (EUS) and EUS-guided fine-needle aspiration (EUS-FNA) with cyst fluid analysis and cytology [11,12,13], misdiagnosis is still common among resected lesions [14]. As a consequence, unnecessary surgery rate is still high, carrying a significant risk of morbidity and mortality, even in expert centers [15].

New EUS-based innovations have been developed to overcome these limitations. In particular, EUS-guided through needle biopsy (EUS-TTNB) has gained interest in the past few years as an important tool for cyst type determination and neoplastic cyst risk stratification [16,17,18]. Several systematic reviews and meta-analyses revealed higher diagnostic performance of EUS-TTNB when compared with EUS-FNA [19,20,21,22,23]. However, safety concerns [24] and questions regarding patient selection [25,26], as well as doubts about the procedure’s clinical impact [27,28], have prevented its widespread adoption among endoscopists.

In this review, we aimed to summarize several aspects regarding using EUS-TTNB to evaluate PCLs, namely the available devices, indications, technical tips, specimen handling, and adverse events. The literature search was conducted using Medline and Embase electronic databases until April 2024. Text words and MeSH terms (pancreatic cyst, pancreatic intraductal neoplasm, cystadenoma, mucinous, endoscopic-ultrasound-guided fine needle aspiration) were used in combination with through-the-needle biopsy, EUS-guided microforceps biopsy, and Moray microforceps. Cross-references were identified manually through the citation list of selected articles to capture additional sources.

## 2. Devices

The first two cases of EUS-TTNB of PCLs were described in 2010 by Aparicio et al. in a pilot study using a 0.8 mm diameter forceps (Lumenis Surgical, Santa Clara, CA, USA) [29]. More recently, the Moray™ microforceps (Steris, Mentor, OH, USA) was developed specifically to target pancreatic cysts [30]. The sheath diameter is 0.8 mm, and the opening width of its toothed jaws is 4.3 mm. Its use has been described in several, mostly retrospective, observational studies [31,32,33,34,35,36,37,38,39,40,41,42,43,44,45]. A third new microforceps, the Micro Bite™ (MTW Endoskopie Manufakture, Wesel, Germany), was developed and used in 25 EUS-TTNB procedures by Stigliano et al. in a retrospective study published in 2021 [46]. The shape of the forceps is oval and toothed with a spoon-shaped mouth and a diameter of 0.8 mm (Figure 1).

These devices are passed through the lumen of a standard 19-gauge FNA needle, allowing the sampling of cyst wall, septa, or mural nodules for histologic analysis of cyst epithelium and stroma [16,38,47].

## 3. Indications to EUS-TTNB and Patient Selection

Computed tomography and magnetic resonance imaging are imperfect for cyst-type determination and differentiation [9,10,48]. Moreover, EUS-FNA cytology is hindered by the low cellularity of the samples [49,50,51], and cyst fluid CEA levels were reported to have sub-optimal accuracy for mucinous cyst diagnosis and do not correlate with the grade of dysplasia [52,53].

EUS-TTNB adoption is far from widespread [54], and indications of its use are being refined [25,27]. The most consensual indication is for the morphologically indeterminate cystic lesion when its nature cannot be determined by imaging and EUS-FNA (cytology and cyst fluid chemistry). Determining specific cyst types frequently has important implications for patient management. Indeed, the treatment and follow-up of many different cysts are based on the histologic subtype, regardless of the presence of worrisome features. EUS-TTNB has been shown to allow accurate preoperative diagnosis of SCA, solid pseudopapillary neoplasia, cNETs [55], MCN [40,56], and other rare cysts [57,58].

As per international evidence-based guidelines, IPMNs with worrisome features indicate EUS with or without cyst sampling [6,7,8]. For mucinous cysts, especially IPMNs, which comprise the majority of incidental pancreatic cysts, EUS-TTNB may be used for risk stratification, as clinical and morphological risk factors in the form of worrisome features, are associated with unnecessary surgery in a significant number of patients [59,60]. In fact, EUS-TTNB allows the determination of the grade of dysplasia in mucinous lesions with improved detection of advanced neoplasia [40,42] and good concordance with surgical specimens for the histologic grade [20,21]. However, clinicians must be aware of the possibility of inhomogeneous distribution of dysplasia within the same lesion, which may result in sampling error. Moreover, the EUS-TTNB trajectory is limited by the FNA needle entry point in the cyst [34], and sometimes, it is impossible to target the most suspected region of the cysts.

EUS-TTNB has also shown in several studies to allow preoperative IPMN subtyping based on histologic morphology and mucin expression [32,42,45]. This issue could be very important for decision-making and risk stratification [25] since phenotype was recently associated with the natural history and malignancy risk in IPMNs [61]. However, considering the intracystic variability of IPMN subtypes [62], further studies should assess the reliability of IPMN subtyping and expression of mucins on TTNB samples compared with surgical pathology as well as the interobserver agreement among pathologists was found to be moderate for IPMN subtyping on surgical specimens [63].

Several authors reporting on the molecular analysis of pancreatic cyst fluid identified distinct mutational profiles for different PCLs, and recently, Paniccia et al. also shown prospectively that for mucinous cysts, next-generation sequencing analysis (NGS) allows the determination of risk for advanced neoplasia [64].

In addition to cyst fluid molecular analysis, it has also been demonstrated that EUS-TTNB specimens are also suitable for NGS in a small series of 23 specimens published in 2019 [65]. The same group recently published a larger study, including 91 specimens submitted to NGS. Adequacy for NGS was 82.4%. Sensitivity and specificity values of 83.7% and 81.8%, respectively, were demonstrated as for diagnosing a mucinous cyst and 87.2% and 84.6% for diagnosing an IPMN. Therefore, these preliminary data suggest that EUS-TTNB could be used for combined histologic and molecular diagnosis, with a potential advantage for risk stratification [66].

Overall, EUS-TTNB is especially useful when a definitive diagnosis of the nature of the cyst is critical to determine the patient’s management, with the ultimate goal of reducing the rate of inappropriate resections or modifying the follow-up protocol [25]. However, to obtain a satisfying result, patient selection is crucial. In fact, Kovacevic et al. performed a prospective study on 101 unselected patients (all patients with a cyst larger than 15 mm were enrolled) [42]. They reported a clinical impact of 12% (10 follow-up discontinuation and two mucinous cyst diagnoses) at the cost of a 10% adverse event rate [42]. EUS-TTNB offers the chance to evaluate a micro histological sample of the cyst wall. Therefore, the greater advantage of the procedure is gained when the histotype of the cyst is unknown, and the “name” of the cyst changes its management. For example, MCN or cNET is almost always indicated for resection, whereas SCA is not. This situation typically happens in the case of unilocular/oligocystic PCL. This cyst “morphology” encompasses numerous cysts, including IPMN, MCN, SCA, or rare conditions, such as cNET [55] or schwannomas [67,68]. Therefore, EUS-TTNB should be used to answer the question “What is this cyst?”, but only if the answer can modify the decision-making process [25]. Indications and contraindications of the technique are summarized in Table 1.

## 4. Technique

The EUS-TTNB procedure is not standardized, contributing to the high heterogeneity among the included studies in two recently published meta-analyses [19,20]. Unfortunately, a few studies have included a detailed description of the technical steps of the procedure, and several technique variations have been described [69]. In general, the cyst is punctured with the needle, the microforceps are advanced into the cyst, and the jaws are opened by the assistant in a similar fashion to standard endoscopic forceps. Then, the forceps are gently pushed against the cyst wall and closed. It is suggested to wait a few seconds before retrieving the forceps to guarantee complete closure of the jaws. If the cyst wall has been properly secured, the wall is retracted toward the needle tip, and the so-called “tent sign” is observed [34]. Observed “tenting” of the cyst wall is a good predictor of an adequate sample. In most cases, it is possible to withdraw the micro forceps from the needle, keeping the latter in the cyst. However, if the volume of the grabbed specimen is too large, passing the forceps jaws backward into the needle is impossible. Therefore, retrieving the needle and the micro forceps together may be necessary. Alternatively, the cyst wall may be released, and another bite is performed (Figure 2).

The technical success, measured by the ability to obtain a macroscopically visible specimen, is reported to be very high [31,32,33,34,35,36,37,38,39,40,41,42,43,44,45,46]. The small number of technical failures was associated with unfavorable scope positions that may be overcome by using more flexible needles made of nitinol [69].

Most of the variations in relation to the technique concern timing for cyst fluid aspiration, forceps preloading, number of passes, bites per pass, number of collected specimens, and specimen handling [31,32,33,34,35,36,37,38,39,40,41,42,43,44,45,46].

The majority of the authors describe performing TTNB before cyst fluid aspiration [40,42,45,46], and some reports describe the possibility of performing biopsies with a collapsed cyst after aspiration [30]. A third possibility is to partially aspirate the cyst, perform TTNB, and finally, completely aspirate the cyst [36,38]. To safely perform TTNB, one should ensure enough space for forceps manipulation inside the cyst and that the accessory is visible on the ultrasound image throughout the procedure. This may justify the fact that most experts choose to perform cyst fluid aspiration after biopsies. In fact, the complete aspiration of the cyst before TTNB may require saline injection through the needle for cyst re-expansion. A potential but unexplored advantage of performing TTNB before fluid aspiration is the chance that forceps manipulation results in cell exfoliation into the fluid, leading to a higher sensitivity of fluid cytology [45]. On the other hand, TTNB often causes intracystic bleeding that could contaminate the fluid for biochemical and molecular diagnostics or even result in the impossibility of aspirating the cyst. Therefore, it seems that the initial partial aspiration of the cyst can ensure some “clean” fluid to analyze while completing the aspiration of the cyst after TTNB provides fluid for cytology. Another advantage of performing partial aspiration of the cyst before TTNB is that it reduces the tension of the cystic wall, favoring the grab of the tissue by the forceps’ jaws (Figure 3).

Forceps preloading in the needle, keeping the distal end of the forceps a few mm inside the needle tip, may avoid cyst wall collapse after the puncture, which would reduce the space for its manipulation inside the lesion, especially in smaller lesions, due to the fluid suction into the needle lumen after stylet removal [32,41,45,46]. A possible disadvantage of forceps preloading is that it reduces the needle/forceps bundle flexibility, which could make the cyst puncture more difficult. On the other hand, when the scope is torqued, as occurs in the second part of the duodenum, it may be difficult to advance the micro forceps through the needle due to resistance from within a bent needle. If resistance is experienced, a slightly gentle movement of opening and closing the forceps may reduce the friction and help overcome this issue. Still, this issue needs further evaluation in future studies.

Regarding the number of passes, a systematic review described that a mean of 3.1 forceps passes is needed to obtain adequate histologic samples [19]. Still, more importantly, we should guarantee the acquisition of two fragments with a minimum number of forceps passes since, as described in the study by Crinò et al., two macroscopically visible specimens are enough to guarantee an adequate histological evaluation and a greater number of passes may increase the risk of adverse events [38]. Nevertheless, this topic should be further explored because a more recent study found that a smaller (<4 vs. ≥4) number of visible specimens is associated with diagnostic failure [44]. In contrast, other reports failed to show a significant correlation between the number of specimens and histological diagnosis adequacy [45,46]. These data should also be interpreted taking into account specimen handling and processing and pathologist experience, which may impact diagnostic yield [70]. Additionally, denuded epithelium in the cyst wall may justify difficulties in obtaining adequate samples even with many passes or fragments [47,56].

The number of forceps passes and acquired specimens should be systematically included in the EUS report for future studies to further explore the correlation between specimen number and diagnostic accuracy and between the number of passes and adverse event risk.

Finally, the number of bites per pass remains unstandardized. Most authors performed one bite for each pass [33,34,38,39,41,44,46], whereas others performed two to three bites [36,40,43]. In our experience, the most important predictor of an adequate specimen is the “tent” sign described above. Table 2 summarizes the technical aspects of the TTNB procedure described in the most relevant studies.

## 5. Specimen Handling and Processing

The specimens obtained during TTNB are much smaller than fragments gathered using standard endoscopy forceps, thus challenging technicians and pathologists to find the best specimen handling and processing method [70,71]. Most authors describe the TNNB sample processing as routine histologic specimens (formalin fixation, paraffin embedding, section, and staining) [31,38,42,44,71]. Still, it is also possible to prepare a cell block [72,73] similar to cytology samples [34,45].

After the biopsy, the specimen is extracted from the jaws of the forceps using the included extraction pick or a needle with minimal manipulation and fixated in formalin. All the collected specimens are usually put in the same container, but the group of Verona and the Pathologists from Copenhagen suggest handling each specimen individually [70,74]. Crinò et al. also described using a customized paper–tissue complex. Each sample is embedded and trimmed separately, preventing tissue loss during handling and resulting in more homogenous tissue sections (Figure 4) [70]. This suggestion should be useful when handling all microfragments [74], even gathered during other procedures (e.g., cholangioscopy-guided biopsies). Table 3 summarizes the available tissue handling and processing possibilities.

Apart from establishing the number of specimens needed for an adequate diagnosis, in a paper published in 2019, Crinò et al. propose the assessment of the samples for the fulfillment of 4 histologic criteria (presence of cyst lining epithelium, differentiating mucinous from non-mucinous cysts, defining dysplasia grade, provide a specific cyst diagnosis) [38]. The grade of dysplasia is one of the most important features to report as it will impact management decisions [71]. Still, we must acknowledge that the degree of epithelial atypia may vary along the cyst epithelium, and TTNB may result in an underestimation of dysplasia [34,45,46]. Another important feature worth exploring is the fact that TTNB samples are suitable for immunohistochemical staining in the majority of cases [71] and allow for IPMN subclassification based on morphology and mucin expression [42,45]. There are important differences in invasive progression, recurrence risk, and prognosis between IPMN subtypes, so this information could affect decision-making [75,76,77,78].

The material acquired with EUS-TTNB is not easy to retrieve and should be handled carefully to avoid wasting it. As reported by Rift et al., after specimen sections have been obtained from the paraffin block, the first and the last slides should be stained with hematoxylin and eosin [71]. According to the presence of epithelium, stroma, or both, and in relation to the specimen’s morphology, additional immunohistochemical analyses are performed to assess the final diagnosis [71]. A positive stain in the ovarian-like stroma of estrogen and progesterone receptors indicates the diagnosis of MCN. Positive periodic acid–Schiff (PAS) and inhibin are used to establish the diagnosis of SCA. Furthermore, the subepithelial capillary network can be visualized by staining for CD34 [71]. Importantly, the interobserver agreement among pathologists when describing TTNB specimens was found to be substantial in a study by Larghi et al. regarding specimen adequacy, presence of epithelium and grade of dysplasia, presence of ovarian stroma, and specific diagnosis [79]. Figure 5 shows an example of EUS-TTNB specimen.

## 6. Diagnostic Yield and Clinical Impact

EUS-FNA has a disappointing performance in cyst type determination and malignancy risk stratification in the case of mucinous lesions [51,80], even if fine-needle biopsy needles are used [81,82]. EUS-TTNB revealed a higher diagnostic yield than EUS-FNA for identifying mucinous cysts in several cohort series [31,40,42], confirmed in several systematic reviews and meta-analyses [19,20,21,22,23]. Moreover, the diagnostic performance is not affected by cyst morphology, size, or location [19]. Table 4 summarizes the findings of available meta-analyses [19,20,21,22,23,83,84,85,86,87].

Clinically, the ability to differentiate between specific cyst types has important implications for management. In fact, the diagnosis of SCA avoids the need for surgery or surveillance, except if the lesion is causing symptoms [42], and in the case of MCN, initial surveillance may be acceptable under certain circumstances [88]. EUS provides high-quality images and allows differential diagnosis in case of typical cyst morphology, such as SCA with honeycomb appearance or extrapancreatic lesions mimicking PCLs, such as gastric duplication cyst [89]. However, for MCN, the rate of preoperative misdiagnosis is around 20% based on clinical and imaging modalities [14,48]. Its distinction from IPMNs may be difficult as the connection between the cyst and the main pancreatic duct is sometimes difficult to determine. EUS-TTNB was reported to have a significant clinical impact, allowing the diagnosis of SCA (especially in the case of the diagnostically challenging oligocystic variant), leading to follow-up discontinuation and preoperative MCN detection by documentation of ovarian-type stroma [42,45,47,56].

Recently, in a study published by Chessman et al. that included 44 patients, the authors reported that the diagnostic yield for the combination of the current “composite standard” (morphology, cyst fluid cytology, and chemical analysis) and TTNB was higher (but not statistically significant) than that for each individual modality, which may prove to be clinically useful. The authors concluded that TTNB led to an overall change in clinical management in 39% of cases [37]. These changes included an increase in surveillance discontinuation, a reduction in the number of follow-up imaging and endoscopic studies, and the referral for surgery in 2 out of 28 patients who would have undergone further surveillance as per standard of care [37]. The clinical impact of EUS-TTNB on the management of PCLs should be further evaluated in future prospective studies.

Moreover, the possibility of combining EUS-TTNB with another through-the-needle technique, i.e., needle-based confocal laser endomicroscopy (nCLE) [90], should be considered. In fact, in the above-mentioned study by Cheesman et al. [37], adding EUS-TTNB and nCLE to standard evaluation based on cyst morphology and EUS-FNA increased the diagnostic yield by up to 93% compared to 75% and 84% for EUS-TTNB and nCLE alone, respectively [37]. This increased diagnostic yield reflected a higher clinical impact. Compared to standard evaluation, an overall change in clinical management in 38.6%, 43.2%, and 52.3% of cases was obtained by EUS-TTNB, nCLE, and their combination, respectively [37].

A meta-analysis compared EUS-TTNB with nCLE [86]. Diagnostic yield (defined as the possibility of obtaining a diagnosis) was higher with nCLE (85% vs. 74%, *p* < 0.0001). Still, the sensitivity and the specificity were similar (pooled sensitivity: 80% vs. 86% and pooled specificity: 80% vs. 83% for EUS-TTNB and nCLE, respectively), as were technical success and adverse event rates [86].

Furthermore, a meta-analysis [83] compared EUS-TTNB to molecular analyses of cyst fluid [91]. The diagnostic yield was higher with EUS-TTNB than with molecular analyses (73.1% vs. 54.3%, respectively), but the rates of correctly identified cysts were similar (70.7% vs. 73.1%, respectively). Similarly, the sensitivities and specificities of differentiating benign from low- or high-risk cysts were comparable (sensitivity: 73% vs. 75%; specificity: 88% vs. 72% for EUS-TTNB and molecular analyses, respectively) [83].

Finally, a network meta-analysis compared several EUS-based techniques for diagnosing PCLs, including contrast-enhanced harmonic EUS, fluid biochemical and molecular analyses, nCLE, and EUS-TTNB [92]. nCLE and EUS-TTNB had a significantly higher network ranking of the superiority index than other techniques for differentiating mucinous PCLs. EUS-TTNB also ranked first in identifying malignant PCLs [92].

## 7. Adverse Events

The overall adverse events rate ranges between 8.6% and 10.1% in different meta-analyses [19,20,21,22,23,83,84,85,86,87]. In the meta-analysis of Tacelli et al., the adverse event rate ranged widely from 1% to 23% [20]. In another meta-analysis, the pooled occurrence for intracystic bleeding was 5% (95% CI, 1.2–11.2), and that for acute pancreatitis was 2.3% (95% CI, 0.5–5.3) [21].

McCarty et al. reported a rate of severe adverse events of 1.08% (95% CI, 0.43–2.69) [22].

The most frequent adverse events are acute pancreatitis, bleeding (mainly intracystic), abdominal pain, and infection (Table 5).

In a more recent multicenter retrospective study including more than 500 procedures, the reported overall AE rate was 11.5%, and a model to predict post-TTNB AE was created that included age, number of forceps passes, inability to completely aspirate the cyst, and diagnosis of IPMN [24]. In this study, 15 adverse events were moderate, 9 were severe, and there were 3 fatalities related to 2 acute pancreatitis and 1 sepsis with organ failure [24]. These fatalities were reported in the context of broadened inclusion criteria [25]. Intra-cystic bleeding is the most frequent event, but it is almost always self-limited, requiring no intervention, and should be considered an “incident” according to the ASGE lexicon [93]. As in ERCP [94], periprocedural hydration with Ringer’s lactate and rectal non-steroid anti-inflammatory drugs (NSAIDs) have been suggested to prevent acute pancreatitis, but the multicenter study by Facciorusso et al. [24] and the prospective study by Kovacevic et al. [42] failed to demonstrate a protective effect of rectal NSAIDs or intravenous hydration. However, despite not being statistically significant, Kovacevic at al. observed a reduction in adverse events rate (17.6% vs. 8.3%) after the implementation of perioperative hydration and rectal NSAID administration [24].

Regarding antibiotic prophylaxis, despite the recommendation of current guidelines [95], more recent evidence showed the lack of benefit from antibiotic prophylaxis after EUS-FNA [96,97] and EUS-TTNB [98]. However, the previously cited multicenter study concluded that it might be useful in the subgroup of patients submitted to TTNB with cysts that cannot be completely aspirated [24].

## 8. Limitations of the Study

This review has several limitations that result from the heterogeneity of EUS-TTNB studies, especially regarding the lack of standardization of the technique, the retrospective nature of most of the cited works, as well as their small sample size and follow-up duration. These issues limit the ability to assess long-term outcomes and the true impact of the technique on patient management and prognosis. Addressing these limitations through large-scale prospective studies would enhance the applicability of the findings in future research.

## 9. Conclusions

EUS-TTNB represents a significant advancement in the diagnostic evaluation of PCLs. By providing superior diagnostic accuracy compared to EUS-FNA, it has the potential to reduce unnecessary surgeries and optimize patient management. Despite its promise, EUS-TTNB is associated with a higher rate of adverse events than EUS-FNA. Continued refinement of the procedure and standardizing technique steps are essential to maximizing its benefits and safety. Future studies should focus on investigating technical aspects and on the evaluation of prophylactic measures.

## Figures and Tables

**Figure 1 diagnostics-14-01587-f001:**
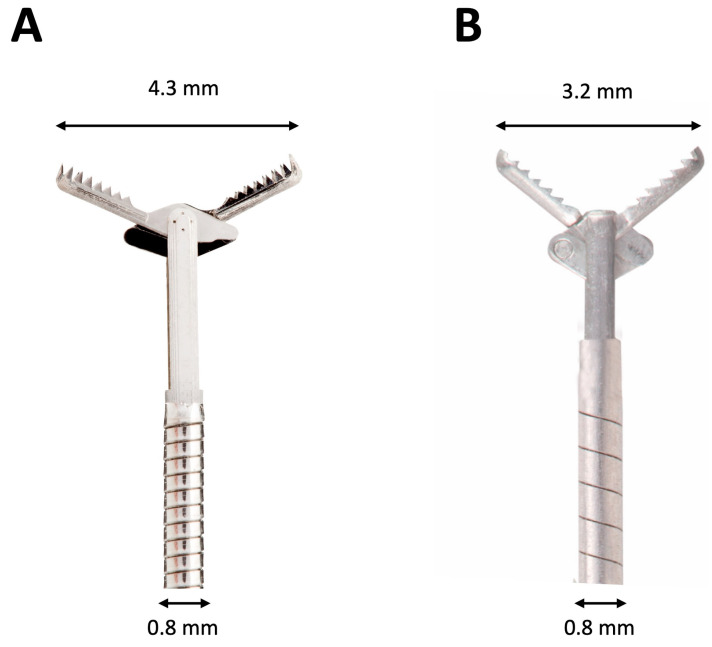
Image depicting key differences between the two currently available microforceps for endoscopic ultrasound through-the-needle biopsy. (**A**) Moray™ microforceps (Steris, Mentor, OH, USA); (**B**) Micro Bite™ microforceps (MTW Endoskopie Manufakture, Wesel, Germany).

**Figure 2 diagnostics-14-01587-f002:**
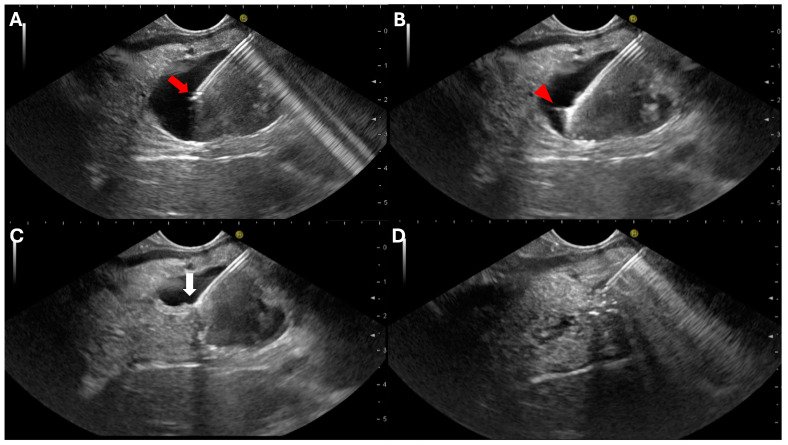
Endoscopic-ultrasound-guided through-the-needle biopsy technique. (**A**) A large unilocular pancreatic cyst is punctured using a 19 G needle (red arrow: needle tip); (**B**) The microforceps is advanced through the needle into the cyst (red arrowhead: microforceps jaws; (**C**) The cyst wall is grabbed by the microforceps jaws and pulled until the “tent sign” (white arrow) is observed; (**D**) The cyst is completely aspirated.

**Figure 3 diagnostics-14-01587-f003:**
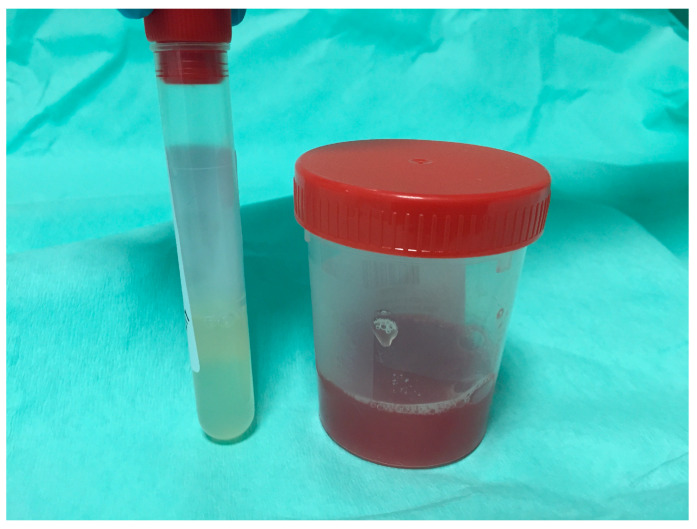
The appearance of cyst fluid aspirated from the same mucinous cyst before (small vial on the left) and after (larger container on the right) endoscopic-ultrasound-guided through the needle biopsy (EUS-TTNB). The fluid aspirated before EUS-TTNB is clean and suitable for biochemical and molecular analyses. After EUS-TTNB, the intracystic bleeding contaminates the fluid that is, however, sent for cytological examination.

**Figure 4 diagnostics-14-01587-f004:**
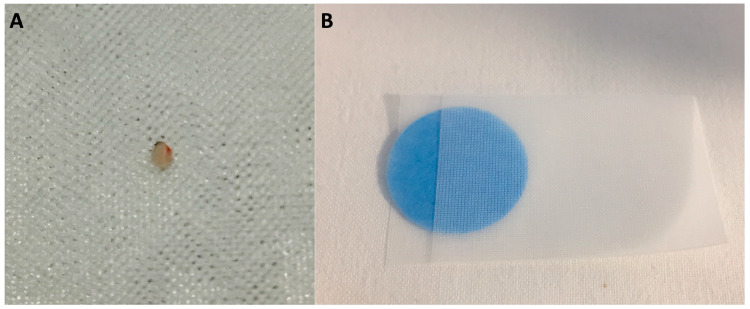
Explanation of the customized handling for specimens collected with endoscopic-ultrasound-guided through-the-needle biopsy. (**A**) The sample is extracted from the microforceps’ jaws. (**B**) The specimen is enclosed between two colored discs of paper, thus creating a “sandwiched” paper–tissue complex that is secured into a gauze envelope and then placed into the formalin container.

**Figure 5 diagnostics-14-01587-f005:**
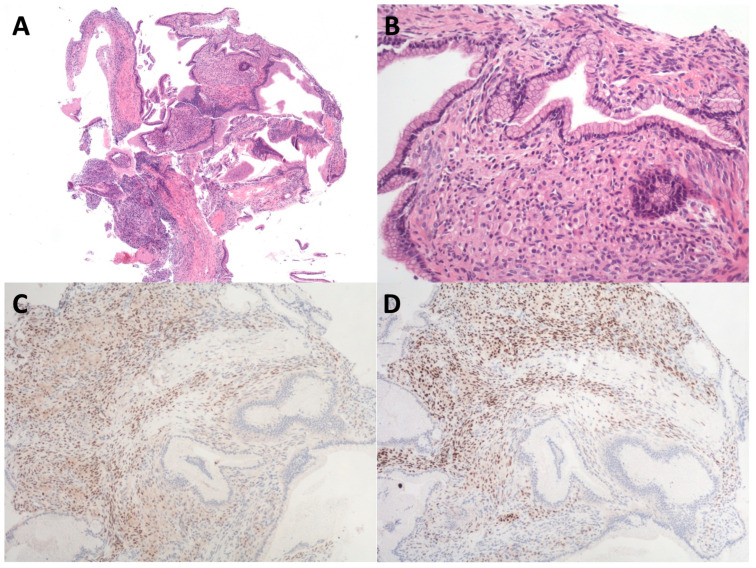
A typical example of histological evaluation of a specimen gathered with endoscopic-ultrasound-guided through-the-needle biopsy. (**A**) The sample is initially stained with hematoxylin and eosin and contains both epithelium and stroma. (**B**) At higher magnification, a single layer of mucinous epithelium covers the highly cellular stroma. (**C**) Stromal cells were estrogen- and (**D**) progesterone-receptor-positive, conclusive for the diagnosis of mucinous cystic neoplasm.

**Table 1 diagnostics-14-01587-t001:** Main indications and contraindications to endoscopic ultrasound through-the-needle biopsy.

EUS-TTNB	Main	Secondary
Indications	Morphologically indeterminate lesion after MRI and EUS-FNA, when knowledge of cyst type may change the management	IPMN risk stratification (grading of dysplasia)
		IPMN subtyping *
		Molecular analysis
Contraindications	When the results will not change patient management	Cyst size < 15 mm Cyst in communication with the main pancreatic duct (e.g., IPMN **)

* Based on morphology and mucin expression. ** Needs proper patient selection as IPMN is associated with a higher adverse event rate. EUS-TTNB, endoscopic-ultrasound-guided through the needle biopsy; MRI, magnetic resonance imaging; EUS-FNA, endoscopic-ultrasound-guided fine-needle aspiration; IPMN, intraductal papillary mucinous neoplasm.

**Table 2 diagnostics-14-01587-t002:** Technical variations in published studies concerning endoscopic-ultrasound-guided through-the-needle biopsy.

First Author, Year	Forceps Preloading	Passes	Bite per Pass	Timing for Cyst Fluid Aspiration	Target Number of Specimen	Antibiotic Prophylaxis
Basar et al., 2018 [31]	No	Until seemed adequate for histologic analysis	N/A	Before	N/A	Yes (broad-spectrum antibiotics, prior to the procedure)
Kovacevic et al., 2018 [32]	No	N/A	N/A	After	N/A	N/A
Mittal et al., 2018 [33]	No	3–4 passes (until 3–4 macroscopically visible specimens)	One	After	3–4 macroscopically visible specimens	Yes (ampicillin-sulbactam or ciprofloxacin, prior to the procedure, continued 3–5 days)
Barresi et al., 2018 [34]	No	At the endoscopist’s discretion	One	Before + after full aspiration	N/A	Yes (piperacillin-tazobactam or ceftriaxone or ciprofloxacin, prior to the procedure, continued 3–5 days post-procedure ^#^)
Zhang et al., 2018 [35]	No	Until seemed macroscopically adequate	N/A	Before	N/A	N/A
Yang et al., 2018 [36]	No	At the endoscopist’s discretion	2–3 bites per pass	Before + after full aspiration	N/A	Yes (following ASGE guidelines)
Cheesman et al., 2019 [37]	No	At the endoscopist’s discretion	At the endoscopist’s discretion	After	N/A	Yes (quinolones *, during the procedure, continued 3–5 days)
Crinò et al., 2019 [38]	No	Until 3 visible specimens are obtained	One	Before	3 visible specimens (each in a different vial)	Yes (ciprofloxacin or piperacillin-tazobactam, prior to the procedure, continued 5 days if intracystic bleeding or cyst not totally drained
Samarasena et al., 2019 [39]	Yes	3–4 passes	One	After	1 adequate specimen	N/A
Yang et al., 2019 [40]	No	3 passes	2–3 bites per pass	Before	1 macroscopically visible specimen	Yes (following ASGE guidelines)
Hashimoto et al., 2019 [41]	No	3–4 passes	One bite per pass	After	N/A	Yes (Cephazolin, prior to the procedure)
Kovacevic et al., 2021 [42]	At the endoscopist’s discretion	Maximum 4 passes	N/A	After	2 macroscopically visible specimens	Yes (Cefuroxime 1500 mg)
Stigliano et al., 2021 [46]	Yes	As required to visible fragment	One bite per pass	After	A visible fragment	Yes (quinolones)
Robles-Medranda et al., 2022 [43]	No	N/A	2–3 bites per pass	N/A	N/A	Yes
Cho et al., 2022 [44]	No	As required to obtain 3–4 visible specimen	One bite per pass	After	3–4 visible fragments	Yes (3rd generation cephalosporins, prior to the procedure)
Vilas-Boas et al., 2022 [45]	Only for lesions ≤ 20 mm	As required to obtain 4 visible specimen	One bite per pass	After	4 visible fragments	Yes (intravenous ciprofloxacin 200 mg, during the procedure)

* Except for allergy (alternative gentamicin). ^#^ Except for one center. N/A, not disclosed.

**Table 3 diagnostics-14-01587-t003:** EUS-TTNB specimen handling and processing.

			Advantages/Disadvantages
TTNB	Specimen handling	Each specimen in individual container and one paraffin block per specimen [70,74]	Avoids tissue loss when cutting because of difference in level between tissue fragments [70].
		All obtained specimens in the same jar.	Reduced processing time.
		Customized paper–tissue complex.	Can be embedded in paraffin to minimize specimen handling; May become the standard for handling all types of small fragments.
	Specimen processing	Same as routine histology specimens [31,38,42,71]	Formalin fixation, paraffin embedding, section and staining. Obtention of several sections for histology and immunohistochemical analysis [71].
		Cell block [34,45]	Similarly to cytology samples. May minimize tissue loss when dealing with microfragments [45,73].

TTNB, through-the-needle biopsy.

**Table 4 diagnostics-14-01587-t004:** Main results on diagnostic yield of published meta-analyses.

First Author, Year	Studies Included	Number of Patients	Technical Success (95%CI)	Diagnostic Outcomes (95%CI)	Comparison with Surgical Pathology
Faias et al., 2019 [83]	TTNB: 4 retrospective Molecular analysis: 2 retrospective 2 prospective	TTNB: 148 Molecular analysis: 1058	TTNB: 93.2% Molecular analysis: 94.9%	Diagnostic yield TTNB: 73.1% (61.4–82.2) Molecular analyses: 54.3% (49.8–58.7) Diagnosis of specific cyst type TTNB: 70.7% (49.4–85.6) Molecular analyses: 73.1% (61.6–82.2)	NA
Facciorusso et al., 2020 [19]	9 retrospective 2 prospective	490	NA	Sample adequacy 85.3% (78.2–92.5) OR vs. FNA cytology: 4.83 (1.63–14.31) Diagnostic accuracy 78.8% (73.4–84.2) OR vs. FNA cytology: 3.44 (1.32–8.96)	Accuracy of 88.3% (80.1–96.5) in patients who underwent resection
Tacelli et al., 2020 [20]	8 retrospective 1 prospective	463	98.5% (97.3–99.6)	Histologic adequacy 86.7% (80.1–93.4) Diagnostic yield TTNB: 69.5% (59.2–79.7) FNA cytology: 28.7% (15.7–41.6)	Concordance with surgical pathology: TTNB: 87.0% FNA cytology: 37.1%
Westerveld et al., 2020 [21]	7 retrospective 1 prospective	426	98.2% (96.8–99.3)	Diagnosis of specific cyst type TTNB: 72.5% (60.6–83.0) FNA cytology: 38.1% (18.0–60.5) OR: 9.37 (5.69–15.42) Diagnosis of mucinous cyst TTNB: 56.2% (45.1–67.0) FNA cytology: 29.5% (15.5– 45.9) OR: 3.86 (2.0–7.44)	Concordance with surgical pathology: TTNB: 82.3% (71.9–90.7) FNA cytology: 26.8% (17.0–37.8) OR: 13.49 (3.49–52.29)
McCarty et al., 2020 [22]	10 retrospective 1 prospective	518	97.1% (93.7–98.7)	Diagnostic yield 79.6% (72.6–85.1) OR vs. FNA cytology: 4.79 (1.52–15.06) Diagnostic accuracy 82.8% (77.8–86.8) OR vs. FNA cytology: 8.69 (1.12–67.1)	NA
Balaban et al., 2020 [84]	8 retrospective 1 prospective	463	98.5% (89–100)	Tissue acquisition 88.2% (79–97) Diagnostic accuracy 68.6% (61%–76)	NA
Guzmán-Calderón et al., 2020 [85]	7 retrospective 1 prospective	423	95.6% (93.2–97.3)	Diagnosis of specific cyst type 74.6% (70.2–78.7)	NA
Rift et al., 2021 [23]	8 retrospective 2 prospective	99	NA	Sensitivity for detection of mucinous cysts TTNB: 86% (62–96) FNA cytology: 46% (35–57) Sensitivity for detection of high-risk cysts TTNB: 78% (61–89) FNA cytology: 38% (23–55) Sensitivity for a specific diagnosis TTNB: 69% (50–83) FNA cytology: 29% (21–39)	All included patients underwent resection
Kovacevic et al., 2021 [86]	TTNB: 9 retrospective 2 prospective nCLE: 2 retrospective 7 prospective	TTNB: 533 nCLE: 557	TTNB: 94% (94–98) nCLE: 99% (97–100)	Diagnostic yield TTNB: 74% (69–78) nCLE: 85% (82–88) Sensitivity for detection of mucinous cysts TTNB: 80% (65–89) nCLE: 86% (69–94)	Concordance with surgical pathology: TTNB: 82% (72–91) nCLE: 65% (36–91)
Gopakumar et al., 2024 [87]	9 retrospective 2 prospective	575	98.6% (97.5–99.4)	Sensitivity 76.6% (72.6–80.3) Specificity 98.9% (93.8–100) Diagnosis of malignant/pre-malignant cyst OR vs. non-malignant: 41.3 (17.4–98.1)	NA

TTNB: through-the-needle biopsy; OR: odds ratio; FNA: fine-needle biopsy; nCLE: needle-based confocal laser endomicroscopy; N/A, not disclosed.

**Table 5 diagnostics-14-01587-t005:** Description of adverse events related to EUS-guided through-the-needle biopsy.

First Author, Year	Overall AEs Rate	Bleeding	Acute Pancreatitis	Infection	Other
Basar et al., 2018 [31]	4.8%	Self-limited intracystic bleeding (*n* = 1, 2.4%)	-	-	Mild abdominal pain (*n* = 1, 2.4%)
Kovacevic et al., 2018 [32]	10.7%	-	Mild (*n* = 2, 7.1%)	-	Non-specific abdominal pain (*n* = 1, 3.6%)
Mittal et al., 2018 [33]	No AEs	-	-	-	-
Barresi et al., 2018 [34]	16.1%	Self-limited intracystic bleeding (*n* = 7, 12.5%)	-	-	Mild abdominal pain (*n* = 2, 3.6%) Moderate-to-severe abdominal pain (*n* = 1, 1.8%)
Yang et al., 2018 [36]	4.2%	Self-limited intracystic bleeding (*n* = 1, 2.1%)	Mild (*n* = 1, 2.1%)	-	-
Cheesman et al., 2020 [37]	9.1%	Self-limited intracystic bleeding (*n* = 1, 2.3%)	-	Moderate-to-severe pseudocyst infection (*n*= 1, 2.3%) *	Mild abdominal pain (*n* = 2, 4.5%)
Crinò et al., 2019 [38]	22.9%	Self-limited intracystic bleeding (*n* = 10, 16.4%) Self-limited peripancreatic bleeding (*n* = 1, 1.6%)	Mild (*n* = 2, 3.3%)	Fever without evidence of infection (*n* = 1, 1.6%)	-
Samarasena et al., 2019 [39]	6.7%	Self-limited intracystic bleeding (*n* = 1, 6.7%)	-	-	-
Yang et al., 2019 [40]	12.3%	Self-limited intracystic bleeding (*n* = 7, 6.1%)	Mild (*n* = 5, 3.5%) Moderate-to-severe (*n* = 1, 0.7%)	-	-
Hashimoto et al., 2019 [41]	3.6%	-	Mild (*n* = 2, 3.6%)	-	-
Kovacevic et al., 2021 [42]	9.9%	Self-limited intracystic on intraductal bleeding (*n* = 4, 4%) ° Mild bleeding in the lesser sac (*n* = 1, 1%)	Mild-to-severe (*n* = 8, 8%) Fatal (*n* = 1, 1%)	-	-
Robles-Medranda et al., 2022 [43]	No AEs	-	-	-	-
Cho et al., 2022 [44]	6.7%	Self-limited intracystic bleeding (*n* = 1, 2.2%)	Mild acute pancreatitis (*n* = 2, 4.4%)	-	-
Vilas-Boas et al., 2022 [45]	7.5%	Mild, self-limited intracystic bleeding (*n* = 2, 5%)	-	-	Mild abdominal pain (*n* = 1, 2.5%)
Stigliano et al., 2021 [46]	10.2%	-	*n* = 3 (6.1%) Mild acute pancreatitis (*n* = 2, 4.1%) Moderate-to-severe acute pancreatitis (*n* = 1, 2.0%)	Infection (*n* = 2, 4.1%)	-
Facciorrusso et al., 2022 [24]	11.5%	Self-limited intracystic bleeding (*n* = 10, 2.0%)	Mild acute pancreatitis (*n* = 9, 1.8%) Moderate-to-severe acute pancreatitis (*n* = 18, 3.6%), Fatal acute pancreatitis (*n* = 2, 0.4%)	Mild Infection (*n* = 5, 1.0%) Moderate-to-severe Infection (*n* = 4, 0.8%) Fatal Infection (*n* = 1, 0.2%)	Mild abdominal pain (*n* = 6, 1.2%) Peripancreatic collection without evidence of pancreatitis (*n* = 1, 0.2%) Hypotension (*n* = 1 (0.2%) *Mass forming* Xanthogranuloma (*n* = 1, 0.2%)

The severity of adverse events (AEs) was graded according to the American Society for Gastrointestinal Endoscopy (ASGE) lexicon [93]. * Post-procedural admission; ° considered as incident according to the American Society for Gastrointestinal Endoscopy (ASGE) lexicon.

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
