# Peer review of "Endoscopic Ultrasound-Guided Through-the-Needle Biopsy: A Narrative Review of the Technique and Its Emerging Role in Pancreatic Cyst Diagnosis"

_diagnostics, 2024, doi:10.3390/diagnostics14151587_

Round 1

Reviewer 1 Report

Comments and Suggestions for Authors

The authors are making a comprehensive review of the use of endoscopic ultrasound-guided through-the-needle biopsy for pancreatic cystic lesions.  They are presenting the technical aspects, the available device, indications, the technique and the results. They compared the results of different alternative techniques like fine needle aspiration and concluded that this procedure is superior in terms of diagnostic accuracy. This procedure is of great clinical importance   mainly because it might spare the patients from unnecessary extensive surgical procedures. However, there is still much to learn about the procedure; the complication rate is high and there is the need for further refinement and development of the technique. The article is well prepared and structured. I have no issues to declare.

Author Response

We thank the Reviewer very much for the positive comment. Below is our point-by-point response:

  1. The authors are making a comprehensive review of the use of endoscopic ultrasound-guided through-the-needle biopsy for pancreatic cystic lesions.  They are presenting the technical aspects, the available device, indications, the technique and the results. They compared the results of different alternative techniques like fine needle aspiration and concluded that this procedure is superior in terms of diagnostic accuracy. This procedure is of great clinical importance mainly because it might spare the patients from unnecessary extensive surgical procedures. However, there is still much to learn about the procedure; the complication rate is high and there is the need for further refinement and development of the technique. The article is well prepared and structured. I have no issues to declare.

Thank you very much for your comment!

Reviewer 2 Report

Comments and Suggestions for Authors

This is a review of EUS-TTNB for pancreatic cystic lesions. 

1. There are many published meta-analyses as cited in the paper. The tittle should be changed to a narrative review.

2. the diagnostic yield of EUS-FNA or TTNB can be mucinous vs. non mucinous, specific type of cysts and degree of dysplasia. This should clearly shown in the desription of study results to avoid confusion.

3. Indications and contraindications can be summarized in a table.

4. A combination of gene alterations can differentiate types of cysts and dysplasia by EUS-FNA (PMID: 36209796). Please discuss.

5. The authors need to emphasize the incremental increase in the diagnostic yield by EUS-TTNB given the increase in cost as well as potential risks of AEs..  

Author Response

The Authors thank the reviewer for their insightful comments. Below is our point-by-point response. 

    1. There are many published meta-analyses as cited in the paper. The tittle should be changed to a narrative review.

    We thank the reviewer for this comment. We changed the title as suggested.

    1. the diagnostic yield of EUS-FNA or TTNB can be mucinous vs. non mucinous, specific type of cysts and degree of dysplasia. This should clearly shown in the description of study results to avoid confusion.

    The authors are thankful for this relevant comment. Indeed, as suggested by the reviewer, the determination of the diagnostic yield for the techniques used to study pancreatic cystic lesions can be described regarding cyst type definition, determine mucinous nature and establish the degree of dysplasia. Contrary to TTNB, FNA does not allow the differential diagnosis between IPMN and MCN, as it does not allow us to obtain ovarian type stroma. Table 4 specifies the rates of different diagnostic outcomes of the several published metanalysis. The majority of the metanalyses determined the diagnostic yield, defined as the proportion of cysts in which a histopathological diagnosis was obtained out of the total number of procedures. Additionally, some studies also described the rate of diagnosis of specific cyst type (Faias, 2019, Westerveld, 2020, Guzmán-Calderón, 2020, Rift, 2021) or the sensitivity for mucinous cyst detection (Westerveld, 2020, Rift, 2021, Kovacevic, 2021). In “chapter 6”, when referring to studies, we specified the different metrics used by the authors (please refer to Chapter 6, paragraph 3).

     Indications and contraindications can be summarized in a table.

    We included a new table summarizing EUS-TTNB indications and contraindications, as suggested by the reviewer (please see table 1, page 4)

    1. A combination of gene alterations can differentiate types of cysts and dysplasia by EUS-FNA (PMID: 36209796). Please discuss.

    The authors thank the reviewer for the suggestion. In the original version of the manuscript Chapter 3 (“Indications to EUS-TTNB and patient selection”), we already included a discussion regarding recent studies describing the relevance of molecular analysis, specifically using TTNB specimens. We now included also the reference to a recent study by Paniccia et al., which used a Next-generation sequencing panel including 22 gene (PancreaSeq) for genomic analysis of pancreatic cyst fluid. This study demonstrated the potential of gene sequencing techniques for the determination of the specific cyst type and the determination of risk for advanced neoplasia. Please see Chapter 3, paragraph 5.

    1. The authors need to emphasize the incremental increase in the diagnostic yield by EUS-TTNB given the increase in cost as well as potential risks of AEs.

    The authors are appreciated by this important remark. In chapter 6 (Diagnostic yield and clinical impact), we thoroughly discuss the incremental value of TTNB for mucinous cyst identification and risk stratification. In table 4 of the updated manuscript we report the diagnostic outcomes described for TTNB compared to EUS-guided FNA cytology.

Reviewer 3 Report

Comments and Suggestions for Authors

The review "Endoscopic ultrasound-guided through-the-needle biopsy: a comprehensive review of the technique and its emerging role in pancreatic cyst diagnosis" provides an excellent summary of information related to EUS in the diagnosis of pancreatic cysts. Recommendations:

1.       The introduction should contain more information about pancreatic cysts.

2.       Add a materials and methods chapter after the introduction to summarize how you selected the materials from the literature.

3.       The indications in chapter 3 should be structured more clearly.

4.       Specify the source of the images.

5.       Tables should only include citations, not the authors' names.

6.       Remove the word "recent" from line 206.

7.       Chapter 6 should emphasize the differential diagnosis that can initially be performed by imaging with EUS, prior to pathological anatomy. I recommend adding the differential diagnosis for rarer pathologies such as gastric duplication cysts, which should be excluded - https://doi.org/10.3390/diagnostics14070675.

8.       The conclusions chapter should be rewritten more concisely.

9.       Add a separate chapter on the limitations of the study.

Author Response

The Authors thank the reviewers for their insightful comments. Below is our point-by-point response:

  1. The introduction should contain more information about pancreatic cysts.

We thank the reviewer for the comment. This narrative review aimed to provide a comprehensive overlook on the technical features, performance metrics and safety aspects of EUS-guided through-the-needle biopsy for the diagnosis and characterization of pancreatic cystic lesions. In the original version of the manuscript, the authors have focused on the technique. In the revised version of the manuscript, we added an extra paragraph to the introduction with information on pancreatic cystic lesions to provide context for the subsequent discussion. Please refer to the section Introduction, paragraph 2.

  1. Add a materials and methods chapter after the introduction to summarize how you selected the materials from the literature.

The authors are grateful for this relevant suggestion. As suggested by the reviewer, we added a section describing the methods for the literature search. Please refer to the section Introduction, paragraph 4.

  1. The indications in chapter 3 should be structured more clearly.

The authors are thankful for this comment. The authors have created a reader-friendly table to summarize the indications and contraindications for EUS-guided through-the-needle biopsy, thus improving the conciseness of the text. Please refer to Table 1 (revised version).

  1. Specify the source of the images.

Figure 1 was prepared using photographs from both forceps and adding the specifications, namely sheath and jaw diameter as well as opening size.

-     Figures 2, 3, 4, and 5 were prepared from the corresponding author´s personal collection.

  1. Tables should only include citations, not the authors' names.

Thank you for your comment. However, per journal policy, Tables reporting data from other studies should indicate the First Author's name, the year of publication, and the reference number. For clarity, we specified “First Author” in the first line of the tables.

  1. Remove the word "recent" from line 206.

As suggested by the reviewer, we agree to remove the word “recent”, as the cited systematic review was published in 2020. This modification can be seen in Chapter 4 (“Technique”), paragraph 6 of the revised version.

  1. Chapter 6 should emphasize the differential diagnosis that can initially be performed by imaging with EUS, prior to pathological anatomy. I recommend adding the differential diagnosis for rarer pathologies such as gastric duplication cysts, which should be excluded - https://doi.org/10.3390/diagnostics14070675.

The authors are grateful for the reviewer’s contribution. As suggested by the reviewer, details regarding EUS cyst morphology are the first information to be obtained. Indeed, it may sometimes be enough to determine cyst type, as is the case for the microcystic variant of serous cystadenoma, which produces a honeycomb-like appearance. Following your suggestion, we added a sentence in Chapter 6 and cited the study you have mentioned.  

  1. The conclusions chapter should be rewritten more concisely.

We thank the reviewer for the suggestion. The conclusions have been rewritten in a more concise form. Please refer to the Chapter 9 (“Conclusions”).

  1. Add a separate chapter on the limitations of the study.

The authors are thankful for this relevant insight. The authors agree with the reviewer that a section dedicated to study limitations may be helpful to identify knowledge gaps and direct subsequent studies. We have added Chapter 8 (“Limitations of the study”) in agreement with the reviewer’s suggestion.

Reviewer 4 Report

Comments and Suggestions for Authors

This review article presents the usefulness of EUS-TTBN for cystic lesions of the pancreas. It is a well-written review article. 1. The title is OK since it reflects the whole study of the article. 2. The abstract is fine. 3. Introduction is acceptable. The authors may explain the types of pancreatic cystic lesions and cancer risks more precisely for readers not familiar with pancreatic cystic lesions. 4. The methods are acceptable. However, the authors request a brief article search method. 5. The results of this review article are OK. 6. In the discussion, is the combination diagnosis of EUS FNA and EUS TTBN a choice? 7. As a result, the authors state, "properly utilizing EUS-TTNB necessitates careful patient selection to mitigate risks and enhance its clinical benefit.” Please state the factors of patient selection in the results. 8. The number of references is enough for the review article. It is also updated. 

Comments on the Quality of English Language

Minor English editing is requierd. 

Author Response

The Authors thank the reviewers for their insightful comments. Below is our point-by-point response:

  1. The title is OK since it reflects the whole study of the article.

We thank the reviewer for the comment.

  1. The abstract is fine.

We thank the reviewer for the comment

  1. Introduction is acceptable. The authors may explain the types of pancreatic cystic lesions and cancer risks more precisely for readers not familiar with pancreatic cystic lesions.

The authors are grateful for this relevant comment. Indeed, this comment is in line with those from other reviewers. Thus, we added an extra paragraph to the introduction with information on pancreatic cystic lesions to provide context for the subsequent discussion. Please refer to the section Introduction, paragraph 2.

  1. The methods are acceptable. However, the authors request a brief article search method.

The authors are grateful for this relevant suggestion. As suggested by the reviewer, we added a section describing the methods for the literature search. Please refer to the section Introduction, paragraph 4.

           The results of this review article are OK.

We thank the reviewer for the comment.

  1. In the discussion, is the combination diagnosis of EUS FNA and EUS TTBN a choice?

The authors are grateful for this very pertinent comment. In Chapter 6 (“Diagnostic yield and clinical management”), we cited the study published by Chessman et al., which points to the advantage of combining techniques to improve diagnostic yield. As suggested by the reviewer, we worked on this aspect of the discussion (please refer to Chapter 6, paragraph 3).

  1. As a result, the authors state, "properly utilizing EUS-TTNB necessitates careful patient selection to mitigate risks and enhance its clinical benefit.” Please state the factors of patient selection in the results.

We thank the reviewer for the comment. In fact, patient selection is paramount to ensure safety when using TTNB in clinical practice. This topic has been addressed by the authors in Chapter 7 (“Adverse events”), when discussing adverse events, we cited the multicenter study by Facciorusso et al., published in 2022, in which a model to predict the risk post-TTNB adverse events was developed. It is stated that age, number of passes, IPMN diagnosis and inability to completely aspirate the cyst are risk factors for adverse events (please refer to the Chapter 7, paragraph 2). Models as described in this study provide relevant tools to assist in the selection of patients for this technique.

  1. The number of references is enough for the review article. It is also updated. 

We thank the reviewer for the comment.

Round 2

Reviewer 3 Report

Comments and Suggestions for Authors

The authors implemented all the suggested recommendations.

Reviewer 4 Report

Comments and Suggestions for Authors

The authors have revised their manuscript well, and it has improved much.